# Translating transcriptomic findings from cancer model systems to humans through joint dimension reduction

Brandon A. Price [1,2], J. S. Marron[1,3], Lisle E. Mose[1], Charles M. Perou[1,2] & Joel S. Parker [1,2✉]

Model systems are an essential resource in cancer research. They simulate effects that we can infer into humans, but come at a risk of inaccurately representing human biology. This inaccuracy can lead to inconclusive experiments or misleading results, urging the need for an improved process for translating model system findings into human-relevant data. We present a process for applying joint dimension reduction (jDR) to horizontally integrate gene expression data across model systems and human tumor cohorts. We then use this approach to combine human TCGA gene expression data with data from human cancer cell lines and mouse model tumors. By identifying the aspects of genomic variation joint-acting across cohorts, we demonstrate how predictive modeling and clinical biomarkers from model systems can be improved.

[1] Lineberger Comprehensive Cancer Center, University of North Carolina at Chapel Hill, Chapel Hill, NC, USA. [2] Department of Genetics, University of North Carolina at Chapel Hill, Chapel Hill, NC, USA. [3] Department of Statistics and Operations Research, University of North Carolina at Chapel Hill, Chapel Hill, NC, USA. ✉email: parkerjs@email.unc.edu

In cancer genomics research, model systems such as cell lines and mouse models are used to permit experimentation that is impossible to do in humans. However, a major concern in utilizing these models is the translational barriers that exist when inferring effects from model system to the human in vivo context. These barriers are one of the sources of variation that contribute to ~91% of preclinical drugs failing to reach the market due to low efficacy in humans[1]. This presents a pressing need to better identify the aspects of model system genomics that can be confidently translated to humans.

To address these challenges, the application of large, public repositories of high-dimensional multi-omics data has become an essential resource for translational studies. Specific to cancer biology, The Cancer Genome Atlas (TCGA) and the Cancer Cell Line Encyclopedia (CCLE) provide invaluable multi-omics profiling across thousands of tumor samples and human cancer cell lines respectively[2,3]. Studies have sought to combine the various data types within a single cohort (*vertical integration*) to better understand the interactions between the biological systems and robust subtype tumors[4,5]. While results from these multi-omic integration studies are promising for investigating more complex genetics, in the context of translating model system effects, the inherent differences between the model and human are not considered.

More recently, methods to integrate data across cohorts (*horizontal integration*) have been presented as a means to address these differences. The joint analysis techniques used in these studies come from a variety of statistical and machine learning domains[6,7]. From these, dimension reduction-based methods have been shown to be successful in both clustering and capturing relevant correlation structure[7–9]. Joint dimension reduction (jDR) approaches decompose input data blocks into lower dimensional spaces that minimize redundant variation and tend to eliminate spurious sampling noise[10]. Correlations among the low-dimension representations are then determined using a range of statistical techniques to determine shared components of variation. Generally, any non-shared variation is removed from the downstream analysis as noise. However, jDR methods such as Joint and Individual Variation Explained (JIVE)[11], Multi-study Factor Analysis[12], and Multi-Omic Factor Analysis[13], parse non-shared variation from noise, isolating variation that acts in a specific manner to a cohort. This cohort-specific variation can then be leveraged to allow a more informed analysis for identifying the drivers behind genomic effects.

Specific to translating cell line or mouse variation to human datasets, methods such as Celligner[14], PRECISE[15], and TRANSACT[16] (a nonlinear implementation of PRECISE) have shown success in clustering model systems and tumor biopsy pairings. These methods have successfully shown how integration can be used to generate more informed hypotheses from model system datasets. However, only TRANSACT has demonstrated an improvement in drug response prediction, but these predictive models have not been validated on clinical datasets outside of the integration (i.e., out-of-sample validation). As a result, the ability for jDR to be used for clinical translation remains uncertain.

Here, we present an approach utilizing the jDR technique *Angle-based Joint and Individual Variation Explained* (AJIVE)[17], to identify the shared and individual behavior of genes across model systems and humans. We integrate cell line and mouse gene expression with TCGA to demonstrate how jDR can separate joint-acting and cohort-specific variation, and be employed to isolate cell line and mice gene expression signals pertinent to human biology. Clinical trial cohorts, independent of the integration and training, provide evidence supporting how jDR can significantly improve response prediction from cell lines, in addition to improving biomarker discovery from mouse models.

## Results

### AJIVE integration captures biologically joint-acting and cohort-specific variation.

AJIVE is an extension of JIVE that employs the use of thresholded Singular Value Decomposition (SVD) to create low-dimension approximations of input data matrices (Fig. 1). Then, through utilizing Principal Angle Analysis, identifies low-dimension subspace bases that either share variation structure (Joint) or contains variation that is unique to a specific input matrix (Individual). By reprojecting the original data ($O_A$, $O_B$) through these subspace bases, we can create joint ($J_A$, $J_B$) and individual ($I_A$, $I_B$) representations of the original inputs. These matrices contain only the shared or unique-acting variation, respectively.

To determine if we could discriminate joint and individual gene expression effects, we conducted a gene-wise integration of RNA-seq data from 1102 human primary breast tumor samples from TCGA (TCGA-BRCA) with RNA-seq from 935 cancer cell lines from CCLE. We evaluated how well the datasets represent one another (Fig. 2a) by comparing the decomposition of the original input data ($O_{TCGA-BRCA}$, $O_{CCLE}$) into the resulting joint ($J_{TCGA-BRCA}$, $J_{CCLE}$) and individual ($I_{TCGA-BRCA}$, $I_{CCLE}$) approximation matrices. We calculated the proportion of joint variation in $O_{CCLE}$ to estimate how much of the variation structure can be captured by $O_{TCGA-BRCA}$, thus giving us a measure for how well observations can be translated from cell lines to primary tumors.

We found the majority of variation across all CCLE cell lines to be shared with TCGA-BRCA ($J_{CCLE} = 50.33\%$) with a smaller portion acting specifically to the cell lines ($I_{CCLE} = 12.15\%$). Grouping cell line measures by tissue reveals that breast-tumor-derived cell lines have, on average, the highest proportion of $J_{CCLE}$ relative to $I_{CCLE}$, making them the most well-represented population of cell lines by $O_{TCGA-BRCA}$ according to this proportion metric (Fig. 2b). When integrating CCLE with other TCGA tumor type datasets (i.e., tumors from different anatomic sites), we found this metric to be highly consistent in identifying expected biologically relevant cell lines (Supplementary Fig. S1). Since CCLE consists of clonal tumor cell lines, we examined how tumor purity may affect the amount of captured joint variation in TCGA-BRCA samples (Supplementary Fig. S2). We observed primary tumor purity had a significant correlation to joint variation proportion ($r = 0.42$, $p < 0.001$).

Clustering cell lines and primary tumors using $O$, $J$ (Fig. 2c–f) and $I$ (Supplementary Fig. S3a, b) show how TCGA-BRCA relevant breast cell lines are clearly captured in the joint component. We observed that after AJIVE integration (Fig. 2d), breast cell lines were not only visibly separated from the remaining cell lines, but also defined within TCGA-BRCA molecular subtypes[18] (Fig. 2f). Furthermore, we observed that tumors continued to cluster according to molecular subtype in the individual variation (Supplementary Fig. S3b). This indicates that there are aspects of primary tumor molecular subtype variation that act specific to TCGA-BRCA and are not being captured by the breast cell lines of CCLE. Similar clustering effects were observed when integrating pan-cancer TCGA samples with CCLE (Supplementary Fig. S3c–e).

Since the primary goal of our analysis was improving the translation of gene expression effects coming from model systems to human breast tumor populations, we prioritized maximizing the amount of translatable joint variation from the cell line components. To do so, we reintegrated TCGA-BRCA with just the 50 breast-tumor-derived cell lines from CCLE (CCLE-BRCA, $O_{CCLE-BRCA}$) expecting a stronger joint component. Integrating $O_{TCGA-BRCA}$ with only breast cell lines resulted in more joint variation ($J_{CCLE-BRCA} = 64.68\%$), as well as a significant decrease in cell line individual variation ($I_{CCLE-BRCA} = 4.52\%$, Fig. 3a). Thus, we continued our cell line analysis using the CCLE-BRCA

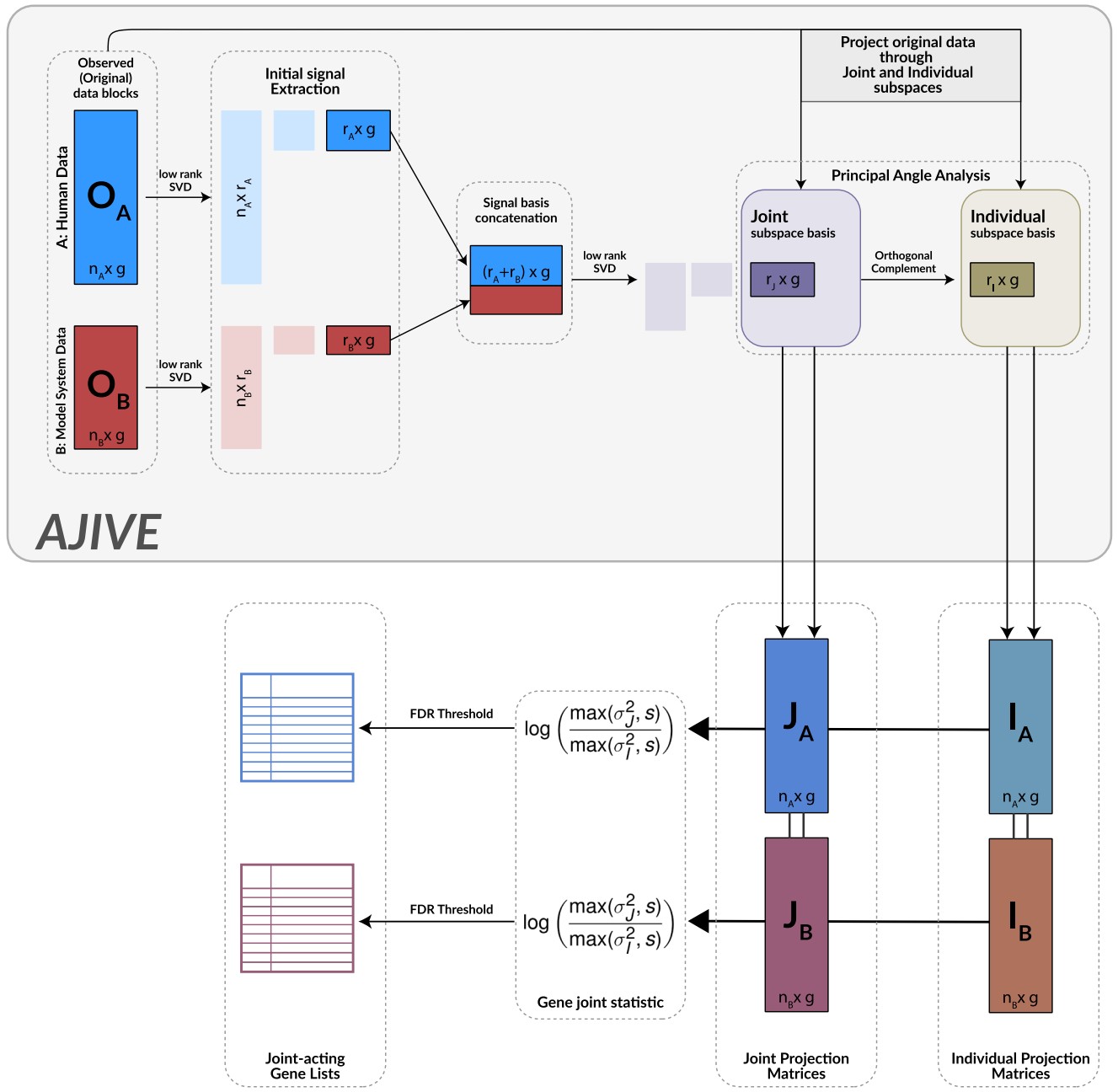

**Fig. 1 Schematic of joint dimension reduction cohort integration using AJIVE.** Human and model system input data are integrated across the gene axis to produce joint and individual projection matrices used in downstream analysis. The original input datasets (**$O_A$, $O_B$**) are decomposed through an initial low-rank SVD using a set initial rank parameter (**$r$**) that removes residual (i.e., noise) variation. Selection of this parameter is detailed in the Methods section. Right singular vectors are then concatenated. Principal angle analysis is performed through an additional SVD step that identifies a "Joint" subspace basis consisting of significant joint-acting components in addition to a corresponding "Individual" subspace basis that consists of input dataset-specific components. The original input data are reprojected through these bases to produce Joint (**$J_A$, $J_B$**) and Individual (**$I_A$, $I_B$**) projection matrices. By calculating a joint statistic for each gene in the projection matrices, we can then identify significantly joint-acting genes across both datasets.

integration; a parallel analysis using the integration with all CCLE cell lines is provided in supplementary information (Supplementary Fig. S4).

In addition, we integrated TCGA-BRCA with a collection of 290 mouse specimens from multiple genetically engineered mouse models (GEMMs, **$O_{GEMM}$**) of mammary cancers[19–21] to compare how well GEMMs represent human tumors against cell lines (Fig. 3b). We observed an increase in $J_{TCGA-BRCA}$ variation (28.60%), but also a significant increase in $I_{GEMM}$ variation (28.82%). This indicates that the GEMMs dataset may be capturing more aspects of variation within TCGA-BRCA.

However, it also suggests that there is a large portion of genomic variation uniquely behaving in mice that, without AJIVE integration, could interfere with an analysis.

We investigated how well-characterized gene expression signatures were being parsed into joint and individual components by observing the variances of genes in **O**, **J**, and **I** from four previously published TCGA-BRCA signatures (one of luminal subtype tumors[22], one of basal-like subtype tumors[23], one of immunoglobulin Bcell/IgG expression[23], and one of general immune-related activity[23]) (Fig. 3c). We expected molecular subtype variation to exist in both breast cell lines and primary breast tumor

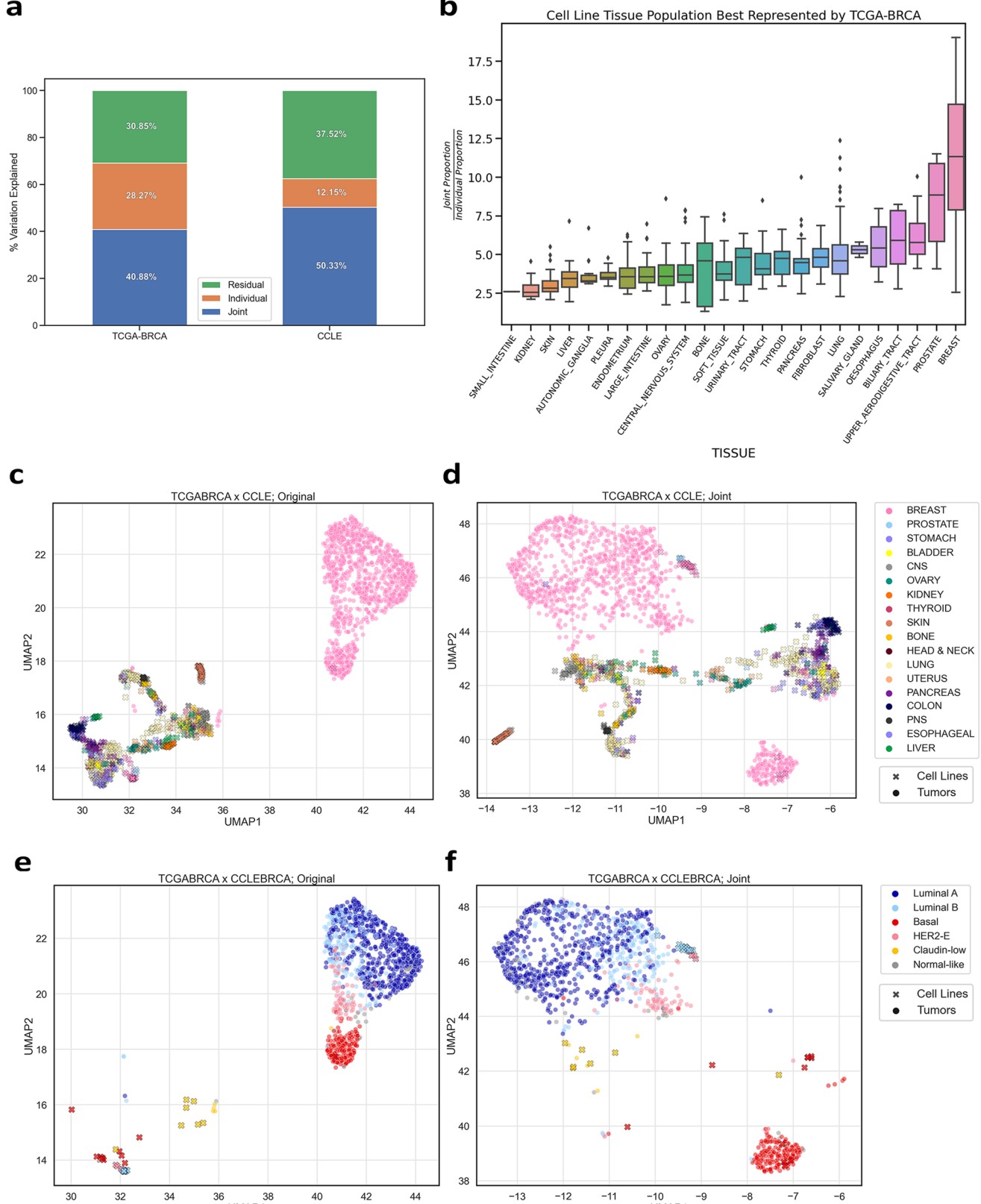

**Fig. 2 Joint dimension reduction integration captures relevant cell lines. a** Percent of variation identified as Joint, Individual, or Residual (Noise) by AJIVE after integrating derived cell lines from CCLE with TCGA-BRCA. **b** Boxplot of the joint-individual proportion ratio for CCLE cell lines grouped by tissue type. High joint proportion and low individual proportion indicates well-represented samples across datasets. 2D projections of combined cell line and breast tumor expression data before integration (Original) and after integration (Joint) using UMAP (*n* = 935 cell lines, *n* = 1102 primary breast tumors). Samples were colored based on tissue type (**c**, **d**) and PAM50 molecular subtype calls[18] (**e**, **f**).

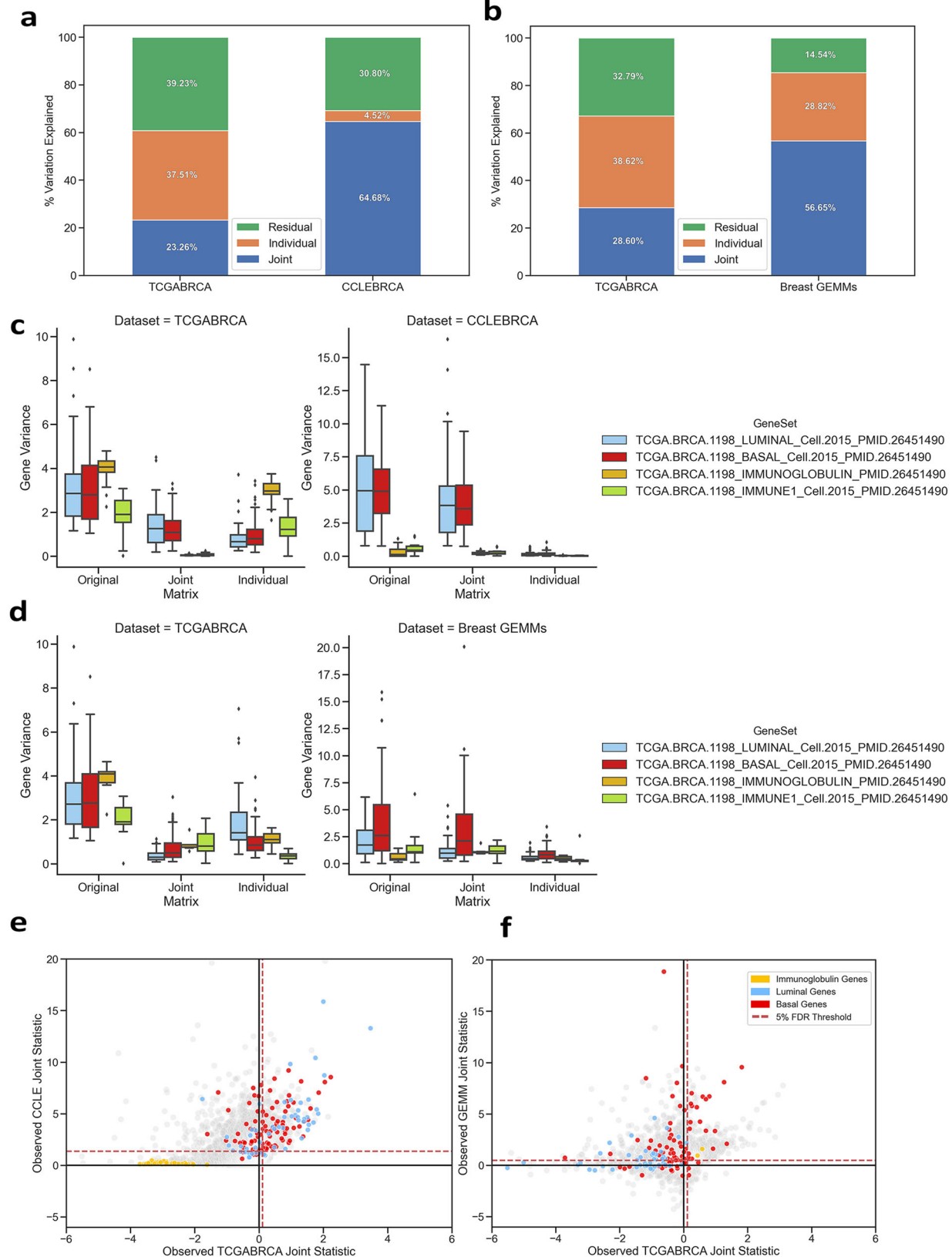

populations[18], therefore be present in the joint component. In contrast, we did not expect gene variation involved in immune response to be in the joint component since $\mathbf{O}_{\text{CCLE-BRCA}}$ does not model the tumor immune microenvironment.

We observed a small fraction of the luminal (3.2%) and basal-like (3.3%) subtype gene variation persisted from $\mathbf{O}_{\text{CCLE-BRCA}}$ to $\mathbf{I}_{\text{CCLE-BRCA}}$ with the majority of variation persisting into $\mathbf{J}_{\text{CCLE-BRCA}}$ (luminal: 84.3%, basal-like: 80.2%). This indicates most the molecular subtype variation in the cell lines is shared with human tumors. However, in TCGA-BRCA, we found a larger proportion of luminal (25.5%) and basal-like (30.6%) variation remained in $\mathbf{I}_{\text{TCGA-BRCA}}$, suggesting there are primary tumor-specific aspects

**Fig. 3 Comparing jDR integrations of breast cell lines and breast GEMMs with TCGA-BRCA.** Integrations were performed between TCGA-BRCA with CCLE-BRCA (**a**, **c**, **e**), and TCGA-BRCA with a collection of GEMMs of mammary cancer[19–21] (**b**, **d**, **f**). **a**, **b** Percent of variation identified as Joint, Individual, or Residual (Noise) by AJIVE after integration. **c**, **d** Boxplots of gene variances of published luminal, basal, immunoglobulin, and immune cell activity gene sets grouped by unintegrated (Original), Joint, and Individual projection matrices. Boxplot elements are defined by upper and lower quartiles, median, and 1.5x interquartile range. Data points outside of 1.5× interquartile range are marked and defined as outliers. **e**, **f** Scatterplots comparing joint statistics for each gene in the integrated datasets. Genes involved in gene sets from **c** and **d** are highlighted. 5% FDR thresholds for each dataset are noted by the red dashed lines.

of subtype variation that the cell lines do not fully capture. Genes involved in immunoglobulin and immune signatures had variation predominantly present in $I_{TCGA-BRCA}$ (74.7%) with only 1.1% of immune-related variation present in $J_{TCGA-BRCA}$, confirming our expectation that immune-related tumor variation would not be captured by the cell lines.

Integrating the TCGA-BRCA data with GEMMs (Fig. 3d), revealed that the immune-related tumor variation was better captured by the GEMMs collection (73.6%). In addition, we found basal-like gene variation to be disproportionately joint-acting compared to luminal variation in TCGA-BRCA (basal-like $J_{TCGA-BRCA}$: 24.0%, luminal $J_{TCGA-BRCA}$: 12.2%). This agrees with previous research finding these GEMMs do not well represent luminal breast tumors[19].

To generalize this approach of determining joint gene expression behavior, we developed a statistic based on the ratio of a gene variance between the joint and individual matrices. The statistic was used to evaluate the proportion between joint and individual gene expression with positive statistics indicating joint behavior and negative indicating individual. Using the permutation approach proposed by ref. [24], we set an FDR threshold to determine significantly joint-behaving genes between the integrated datasets (Fig. 3e, f). If a gene's joint statistic was observed to be greater than both FDR thresholds, it was defined as being fully joint-behaving or translatable between datasets. If a gene's statistic was greater than only one FDR threshold, then this indicates partial joint behavior where the variation seen in one dataset may not be fully represented by the samples in the other. These genes have the potential to be translated unidirectionally.

When integrating TCGA-BRCA with breast cell lines (Fig. 3e), we observed over half of both luminal (31 out of 53 genes, 58.5%) and basal-like (55 out of 94 genes, 58.5%) subtype genes were significantly joint-behaving between both datasets at a 5% FDR threshold. However, when only considering the CCLE FDR threshold, we observed many more subtype genes (91.5% luminal, 81.1% basal) to, at a minimum, be unidirectionally joint-acting (i.e., the variation observed in these cell line genes are fully captured by samples in TCGA-BRCA, but the same genes in TCGA-BRCA are not being completely represented by the cell lines). This agrees with our previous finding that molecular subtype variation is largely shared between cell lines and primary tumors, but there are aspects of subtype variation specific to primary tumors that are not found within the breast cell lines. Furthermore, we found all genes in the previously used immunoglobulin signature to not be joint-acting in either dataset. Conversely, these genes were in the top 1% of negative TCGA-BRCA gene statistics indicating significant primary tumor-specific behavior ($p < 0.001$). Integrating TCGA-BRCA with GEMMs (Fig. 3f) revealed that immune-related homologs in mouse were significantly joint-behaving with TCGA-BRCA. In addition, we observed more basal-like genes (67%) than luminal (43.4%) to be unidirectionally joint-acting with only 19 basal-like genes being significantly translatable between both datasets.

**Utilizing joint variation improves the identification of lapatinib response factors.** By identifying gene variation that is well characterized and common between TCGA-BRCA and CCLE-

BRCA, we anticipated that the joint variation could be utilized to improve the knowledge derived from cell lines. Drug response data of cell lines are known to be highly variable and challenging to train predictive models on due to cell line nuisance variation resulting from both technical and in vitro artifacts[25]. Using AJIVE integration, we explored if this nuisance variation could be mitigated to produce more accurate models of human variation.

To investigate how AJIVE integration captures known genetic associations of drug response, we evaluated how known gene targets of lapatinib, a tyrosine kinase inhibitor of *ERBB2* (*HER2*) and *EGFR*, correlate to the joint, individual, and original versions of the CCLE-BRCA expression data (Fig. 4a). We found both $O_{CCLE-BRCA}$ and $J_{CCLE-BRCA}$ adjusted data matrices to have a significant correlation of *ERBB2* expression with lapatinib sensitivity ($p = 0.009$, $p = 0.02$). However, *ERBB2* had an opposite correlation within $I_{CCLE-BRCA}$ ($p = 0.003$), suggesting potential aspects of cell line-specific variation may mask tumor sample effects. This masking effect can be better seen in our full CCLE analysis (Supplementary Fig. S4D) where the inclusion of non-breast cell lines resulted in *ERBB2* expression having a greater correlation to lapatinib resistance in $I_{CCLE}$. Removal of this variation strengthened the correlation to lapatinib sensitivity in $J_{CCLE}$ ($O_{CCLE}$ vs. $J_{CCLE}$ correlation difference $p$ value $= 0.04$,[26]).

**Improving clinical ERBB2-targeting response prediction through jDR.** Investigation of the joint and individual effects supports the notion that tumor expression signatures and drug response associations are contained within the joint component. We then hypothesized that AJIVE integration could benefit predictive modeling of cell line drug response data by removing nuisance variation. We developed three elastic net models of *ERBB2*-targeting response using the original, joint, and individual approximations of CCLE-BRCA when integrated with TCGA-BRCA.

The elastic net models were applied to breast cancer samples from CALGB 40601 (NCT00770809); this neoadjuvant trial of HER2 + breast cancer encompassed three arms treated with a combination of paclitaxel + trastuzumab, paclitaxel + lapatinib, and paclitaxel + lapatinib + trastuzumab. Measurements of mRNA-seq expression were collected from the 305 patients[27], and pathological complete response (pCR) was used as the primary outcome. Combining all trial arms together, we predicted *ERBB2*-targeting response and pCR status for each sample and generated a receiver operating characteristic (ROC) curve (Fig. 4b). By comparing the mean area under each model's ROC curve, we found training our model using the joint variation significantly improved ($p = 0.03$) our predictive performance over training over the full, unintegrated CCLE-BRCA dataset (Fig. 4c). Examining the model performance by clinical trial arm (Supplementary Fig. S5) revealed improvement occurred in the paclitaxel + trastuzumab (TH) arm and paclitaxel + trastuzumab + lapatinib (THL) arms; we note here that the paclitaxel + lapatinib (TL) arm was stopped early due to lack of efficacy and thus has a smaller sample size than the other arms. To test the extent to which integration improves the success of the models, a retrospective analysis was performed bootstrapping the AJIVE integration and elastic net modeling procedures at varying breast

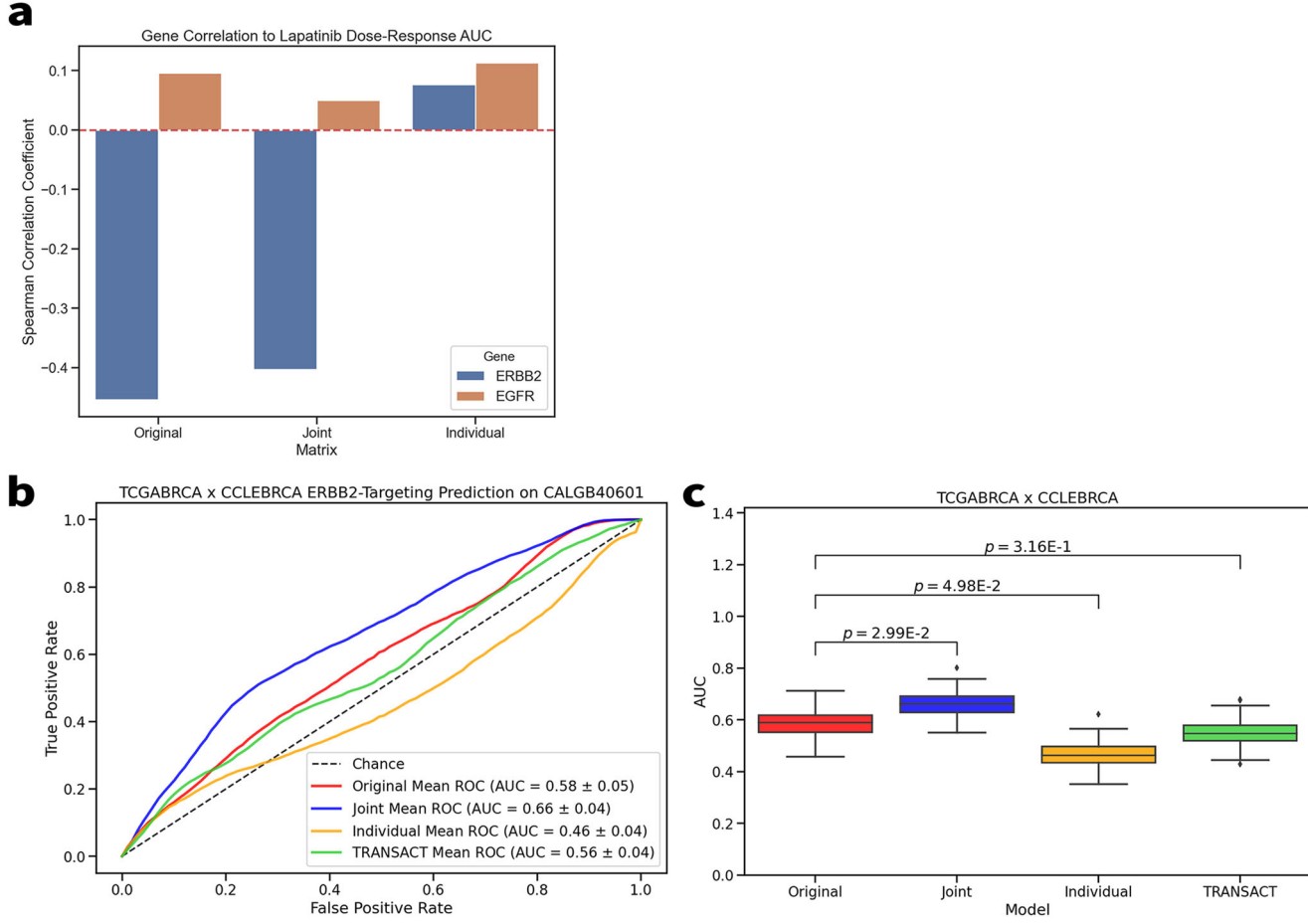

**Fig. 4 Modeling lapatinib response from cell lines using joint and individual matrices. a** Spearman correlation coefficients of ERBB2 (HER2) and EGFR expression to lapatinib dose-response AUC in CCLE-BRCA. **b** Receiver operating characteristic (ROC) curve of elastic net ERBB2-targeting response models trained from Original, Joint, and Individual CCLE-BRCA AJIVE projection matrices in addition to TRANSACT-derived principal vectors. Models were tested on CALGB 40601 using pathological complete response (pCR) as the positive class. **c** Boxplot of ROC-AUCs. Empirical p-values were derived from differences between bootstrapped predictions.

cell line sample sizes (Supplementary Fig. S7). By adding more breast cell lines to the integration, the amount captured joint variation was increased, leading to improved predictive model performance using the joint matrix.

To compare with our AJIVE-based jDR approach, we repeated the experiment using TRANSACT[16]. We found applying the TRANSACT-trained model had no significant change in predictive performance over using the original, unintegrated CCLE-BRCA data (Fig. 4b, c).

**Joint integration captures clinically translatable gene variation in GEMMs.** As with our cell line experiment, we hypothesized that AJIVE integration would provide benefit in translating genomic effects of mice to human clinical data through removing noise and isolating clinically informative variation. In a study by Hollern et al., a large collection of mammary tumor GEMMs was used to discover a B cell/T cell co-cluster signature that was predictive of immune checkpoint inhibitor (ICI) sensitivity in human melanoma patients, and predictive of response to chemotherapy and/or trastuzumab therapy in human breast cancer patients[19]. Given this known translatable biomarker, we assessed the joint behavior of the signature between GEMMs and TCGA-BRCA, and tested if the signature could be further refined through joint integration.

The B cell/T cell co-cluster signature uses a collection of 22 genes. We observed the joint statistic for each of these genes

against the background between TCGA-BRCA and the GEMMs dataset (Fig. 5a). We set an FDR threshold to select significant joint-acting genes between the two datasets. At an FDR threshold of 1%, 16 of the 22 genes in the signature were significantly joint-acting. We tested both the full 22 genes and reduced 16 gene signatures in two breast cancer clinical trials, CALGB 40601 (NCT 00770809)[28] and MADRID (NCT 01560663)[29] (Fig. 5b, c). We found both signatures were equally enriched with patients that responded to therapy. In both trials, the six genes filtered out possessed no informative value between response groups.

In a human melanoma study of anti-CTLA4 therapy[30], the signature was enriched in responsive samples. We built elastic net models of anti-PD1, anti-CTLA4 therapy response from the integrated and unintegrated versions of the GEMM dataset with TCGA-BRCA. These models were then used to predict ICI response in the human melanoma study (Fig. 5d). We found training our models on the AJIVE integrated joint variation to have equivalent performance to using the entirety of the original GEMM dataset (Original ROC-AUC = 0.73 ± 0.06, Joint = 0.76 ± 0.07).

To test how the integrated dataset affects predictive performance, we reintegrated the GEMM dataset with different subtypes of TCGA and retrained our elastic net models. After applying these models to the human melanoma study (Fig. 5e), we determined integration with TCGA-BRCA performed best (Mean AUC = 0.76 ± 0.07) and integration with TCGA-GBM

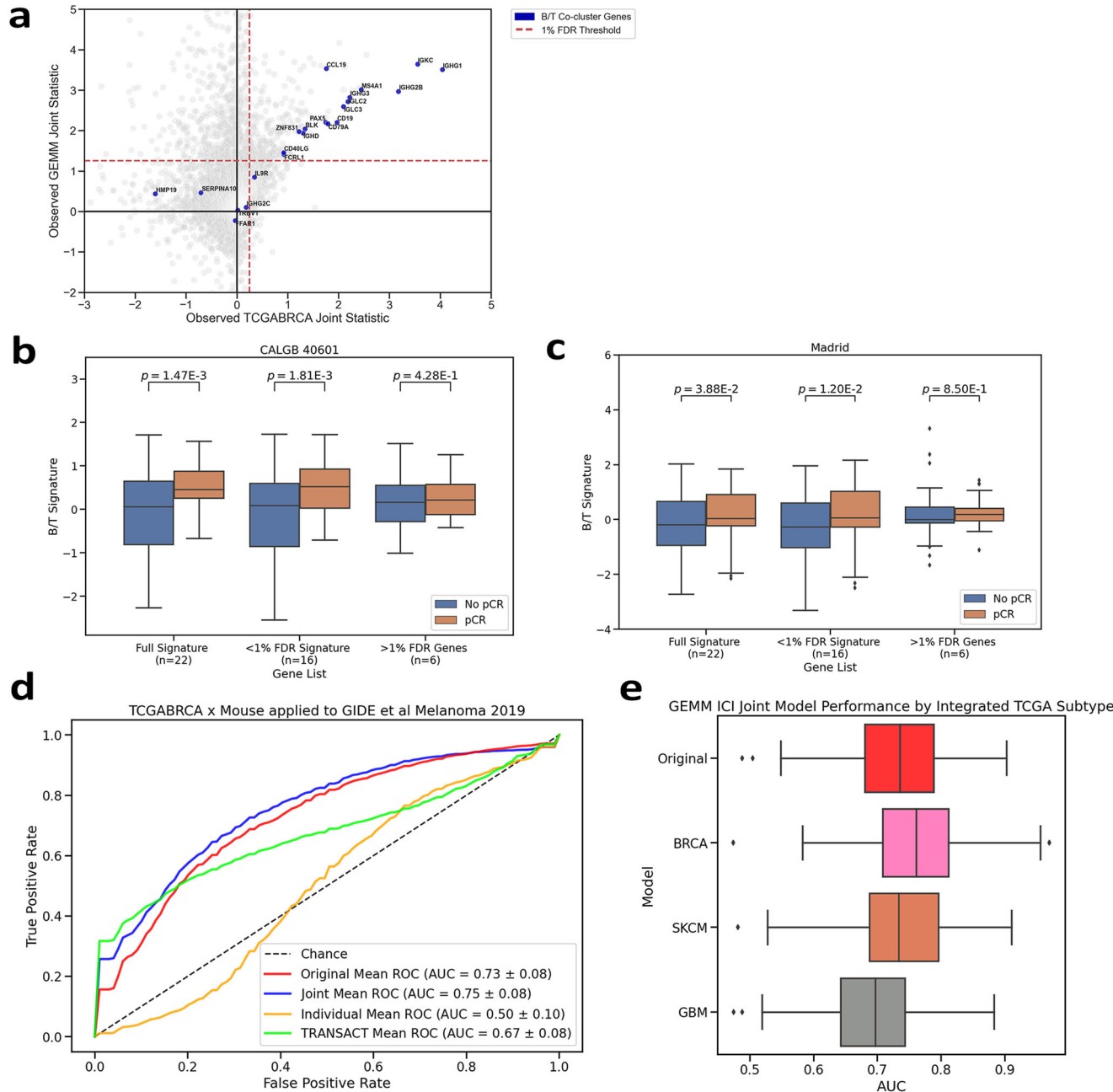

**Fig. 5 Integrating GEMMs data with TCGA-BRCA. a** Scatterplot of joint statistics of each gene in mouse and TCGA-BRCA joint datasets. Genes involved in the B cell/T cell co-cluster signature described in ref. [19] are highlights. Red dashed line indicates 1% FDR threshold of significant joint behavior. **b**, **c** Boxplots of B cell/T cell signature using all genes, significant joint genes, or filtered genes, applied to CALGB 40601, trastuzumab arm[28] and pretreatment samples from the TNBC MADRID NCT 011560663 clinical trial[27–29]. Signature was calculated as median expression of genes. Both datasets split by pCR. Two-tailed *p* values calculated from standard T-tests as in ref. [19]. **d** Receiver operating characteristic (ROC) curve of anti-PD1/anti-CTLA4 response models trained from Original, Joint, and Individual GEMMs data AJIVE projection matrices. Models were tested in Gide et al., 2019 human melanoma study using pathological complete response (pCR) as the positive class[30]. **e** Boxplot describing ROC-AUCs of alternative anti-PD1/anti-CTLA4 response models trained on GEMMs data integrated with other TCGA subtype data.

(Glioblastoma) performed the worst (Mean AUC = 0.68 ± 0.05) compared to the unintegrated model.

## Discussion
Model systems, be they cell lines or animal models, are essential tools for cancer research. However, it is often challenging to translate model system findings into human tumor findings, especially for predicting sensitivities to drugs. Barriers to this translation are many including inherent biological differences,

but often these differences are more technical in nature. To address this challenge, we implemented a jDR integration technique across different RNA-seq datasets, which allowed us to identify shared and non-shared components of gene variation between model systems and humans. The shared, joint-acting, variation can be used to better identify genes that will translate to a target human population. The non-shared, individual-acting, variation can be used to either remove biases from downstream analysis or identify unique aspects of each cohort.

We found analyzing joint variation within genome-wide gene expression datasets lends higher confidence to finding genuine effects occurring in both datasets. This allows predictive models to be better translated from one dataset to another in a more assured manner. When training Elastic Net regression models using joint variation, we found out-of-sample clinical predictive performance to significantly improve when modeling *ERBB2*-targeting response in cell lines (ROC-AUC = 0.66), suggesting drug prediction models could be made more clinically relevant by applying jDR integration with an external dataset that represents the target clinical population. To reinforce this notion, we modeled immunotherapy response from breast GEMMs and applied these models to a human melanoma study. Although performance remained the same after jDR integration with TCGA-BRCA, the choice of integrated dataset had a significant effect on the model's predictive performance and could potentially be improved by selecting datasets that better represent both the model system population and clinical population. In scenarios where representative datasets are not available, our findings using jDR to integrate pan-cancer cell lines from CCLE and all tumor types of TCGA (Supplementary Fig. S4) suggest that jDR integration could be used to identify relevant subpopulations within large public data repositories. Furthermore, we found TCGA-BRCA sample purity correlated with the amount of joint variation, indicating that jDR is capturing the translatable aspects of both high and low-purity samples. As a result, we hypothesize that jDR integration could be used to help remove the confounding effects of purity level from an analysis[31].

The application of horizontal integration presents a promising avenue for training drug response prediction models. Work by Mourragui et al. has demonstrated how different drug prediction modeling strategies could improve training after horizontal integration, with median ROC-AUC's averaging between 0.56 and 0.62 across 17 drugs and 4 modeling strategies[16]. While these performance measures were validated on one of the datasets used in the integration, we have demonstrated successful application of horizontal integrated drug prediction models to out-of-sample clinical trials. This was done using Elastic Net models that could potentially be improved through implementing non-linear modeling strategies as done by Mourragui et al.

The use of jDR integration provides a semi-supervised approach for identifying clinically-translatable gene variation. By creating a statistic to formalize this identification, we demonstrated how genes of clinical importance can be better identified from mouse data. For clinical biomarkers, filtering out untranslatable genes could prove to be immensely beneficial by permitting a more precise measurement or better understanding of the biomarker's biology. In addition, biomarker discovery could be improved by removing genes that would otherwise create noise in an association analysis.

In conclusion, we provide evidence that AJIVE-based jDR integration can be used to improve the translation of cell lines and mouse genomics to human clinical data. Through integration, we have begun to address the potential blind spots in model system research by objectively identifying shared features with human specimens.

## Methods

**Dataset collection and processing**. TCGA-BRCA (https://portal.gdc.cancer.gov/projects/TCGA-BRCA), CCLE (GEO: GSE36139), CALGB 40601 (dbGaP Study Accession: phs001570.v2.p1 and GEO: GSE116335), and (dbGaP Study Accession: phs001427.v2.p1)[30] gene expression data were obtained and processed through an identical pipeline aligning to human reference genome hg38 using STAR and quantified with Salmon. Expression values were upper quantile normalized and log-transformed. Hematopoietic and lymphoid tissue cell lines were removed from CCLE. Genes with no expression in any dataset were removed and an intersecting gene list across datasets was used. Genes were mean-centered as required by AJIVE.

GEMM RNA-seq data were obtained from GEO: GSE124821[19] and was processed as detailed in the article.

Lapatinib response AUC data for the CCLE cell lines were taken from the Cancer Therapeutics Response Portal available at https://ctd2-data.nci.nih.gov/Public/Broad/CTRPv2.1_2016_pub_NatChemBiol_12_109/. In cell lines where multiple AUCs were reported, a mean AUC was used instead.

All analysis and plotting were done in Jupyter using python 3.6.7 with NumPy v1.18.5, SciPy v1.5.2, pandas v1.1.2, matplotlib v3.3.1, seaborn v0.9, and scikit-learn v0.23.2 packages. AJIVE was run using the py_jive package (https://github.com/idc9/py_jive).

**Angle-based Joint and Individual Variation Explained (AJIVE)**. Joint and Individual Variation Explained (JIVE)[11] is a dimension reduction algorithm that extends PCA to multi-block data. Given $K$ data matrices (blocks), JIVE decomposes each matrix into joint (**J**), individual (**I**), and residual noise (**E**) structures:

$$X_k = J_k + I_k + E_k \tag{1}$$

AJIVE extends this approach to utilize singular value decomposition (SVD) and Principal Angle Analysis. An initial SVD step decomposes each matrix into $r_k$ low-rank approximations, i.e., $X_k = U_k D_k V_k^T$ where **U** and **V** are orthogonal across $r_k$ and **D** is a diagonal matrix. Right singular vectors ($V_k$) are then concatenated and principal angle analysis is performed through an additional SVD step:

$$M = \begin{bmatrix} V_1^T \\ \vdots \\ V_k^T \end{bmatrix} = U_M D_M V_M^T \tag{2}$$

where singular values $\sigma_M$, on the diagonal of $D_M$, are used to calculate the principal angles $\phi_{j,k}$ between row ($X_j$) and row ($X_k$):

$$\phi_i = \arccos\left\{ \left(\sigma_{M,i}\right)^2 - 1 \right\} \tag{3}$$

for each $i \in 1, \ldots, r_j \wedge r_k$. Ranks are labeled as joint if the angle is small enough when compared to randomly generated distribution of principal angles within the random subspaces.

**Selection of initial signal ranks**. To determine AJIVE initial rank parameters ($r_k$), we implemented a bootstrap approach on each input dataset to find the optimal ranks at which AJIVE correctly identifies joint variation. Samples within one input data matrix were randomly divided into two equal halves. AJIVE was repeatedly applied on the two halves while increasing the initial rank parameter. Percentage of variation (sum of squares) explained by joint and individual structure were calculated on the output matrices of each rank. Residual variation was defined as any variation not explained by joint or individual structure. The process was repeated ($n = 10$) with joint and individual percentages being collected and plotted (Supplementary Fig. S6). The rank at which joint variation was maximized while minimizing individual variation was selected as the optimal initial rank for each input data matrix.

**AJIVE integration between cancer cell line and human data**. RNA-seq expression data from 935 cell lines from CCLE and 1102 breast tumor biopsies from The Cancer Genome Atlas (TCGA-BRCA) were horizontally integrated across genes using AJIVE[17]. By integrating the datasets across genes, the right singular vectors of the decomposition now represent ranks of "metagenes" across the datasets. Each metagene contains partial combinations of genes that AJIVE then compares and identifies as joint or individually behaving.

For integrating TCGA-BRCA with breast cell lines, we repeated the same integration process using the 50 cell lines from CCLE derived from breast tissue (CCLE-BRCA). AJIVE was run with CCLE and TCGA-BRCA setting initial signal ranks of 150 and 275 respectively. For integrating CCLE-BRCA and TCGA-BRCA, initial signal ranks were set to 35 and 135.

**Sample-wise joint and individual explained variation**. To quantify how the variation of each sample was divided into joint, individual, and residual components, we multiplied the right singular vectors of the final reprojected SVDs of each data block by the diagonal matrix. This maintains orthonormality such that we can calculate percentage of explained variation by the sum of squares with each matrix from the original. We evaluated how well represented a group of cell lines was in an integration (Fig. 2b, Supplementary Fig. S1) by calculating the ratio of joint variation explained by individual variation explained for each cell line and grouping them by tissue.

**Calculating joint statistic**. We developed a statistic to determine the "joint behavior" of a gene. This is represented by the log-ratio of each gene $g$, variance in the joint matrix $\sigma_J^2$ by the variance in the individual matrix $\sigma_I^2$. If the variance of a gene in either joint or individual matrix is not larger than a manually set threshold $s$, then the statistic is set to zero. This threshold is determined through observing the distribution of summed variances and prevents artifacts due to low-variance

genes. For our analysis, $s$ was set to 0.5.

$$\log\left(\frac{\max\left(\sigma^2_{Ig},\, s\right)}{\max\left(\sigma^2_{Ig'},\, s\right)}\right) \qquad (4)$$

$P$ values were computed from an expected distribution through random permutations of combining the joint and individual matrices and shuffling joint/individual labels. FDR values were calculated according to the permutation approach described by ref. [24].

**Lapatinib association testing**. To evaluate if human-relevant aspects of cell line drug response biology can be captured in joint structure, we calculated Spearman rank correlation coefficients of each gene in the original, joint, and individual matrices to lapatinib response AUC (area under the drug response curve) data for the cell lines with response data. In cases where multiple AUC values were provided for a single cell line, the median was used.

**Modeling method and parameters**. We chose to use Elastic Net due to a long history of success in the drug response prediction literature, demonstrating equivalent success to complex models[32–34]. We trained a linear elastic net model on the cell lines that had lapatinib dose-response AUC data from the original, joint, and individual CCLE ($n = 551$) and CCLE-BRCA ($n = 32$) matrices. The following procedure was conducted on each matrix using scikit-learn.ElasticNet parameter definitions:

1. 100 alpha values were calculated at each l1_ratio of 0.1, 0.5, 0.7, 0.9, 0.95, and 1 by computing the elastic net path (sklearn.linear_model.enet_path).
2. We split the data into 20 iterations of training (67%) and test (33%). In CCLE, splits were stratified according to cell line tissue of origin.
3. For each iteration, we fit elastic net models on the training set using an exhaustive grid search over all alpha and l1_ratio combinations.
4. We applied each model to its respective test set and calculated an $R^2$.
5. Final model was selected using the best mean $R^2$ across the 20 iterations.

**Applying models to CALGB 40601**. The final original, joint, and individual elastic net models were applied to CALGB 40601 gene expression data for samples in each arm of the trial. In each arm, a ROC curve was computed using pathological complete response (pCR) as the positive class. Performance was measured by area under the ROC curve (ROC-AUC).

**Comparison with TRANSACT**. TRANSACT[16] is a nonlinear implementation of PRECISE[15] that utilizes kernel-PCA to derive nonlinear principal components that are then geometrically compared into principal vectors. Vectors that best balance the contributions of the input datasets are called *consensus features* and are used to project the datasets into a reduced-dimension joint space. To compare with our AJIVE approach, we repeated our experiment integrating CCLE-BRCA and TCGA-BRCA using TRANSACT and applied the trained predictive model on CALGB 40601. Code to run the integration was used from the TRANSACT package: https://github.com/NKI-CCB/TRANSACT. TRANSACT elastic net models were trained on consensus features using the same modeling procedure as detailed above.

**AJIVE integration between cancer mouse and human data**. Mouse and TCGA-BRCA RNA-seq data were analyzed through an identical AJIVE pipeline using initial rank selections of 60 and 135, respectively. Prior to integration, genes were matched by capitalizing gene labels and translating any homologous genes according to the Jackson Laboratory human-mouse homolog map available at http://www.informatics.jax.org/downloads/reports/HOM_MouseHumanSequence.rpt.

**Mouse predictive modeling**. Predictive models of anti-PD1/anti-CTLA4 response were trained from the GEMMs dataset using logistic regression with elastic net penalty (sklearn.linear_model.SGDClassifier). Four models were trained using the GEMMs samples integrated with TCGA-BRCA, TCGA-SKCM, TCGA-GBM, and the unintegrated GEMMs dataset (Original). An identical cross-validation procedure was implemented as detailed above. Final selected models from cross-validation were tested on a human melanoma immunotherapy study as described in ref. [30].

**Statistics and reproducibility**. All information on the statistical analyses performed in this work has been included in the related figures, figure legends, results, and methods. All statistical tests used to evaluate significance in addition to error bar definitions are described in the figure legends or methods. Statistical calculations were performed using the related functions in the pandas, SciPy and scikit-learn python packages. $R^2$ values were calculated from the linregress function in SciPy.

**Reporting summary**. Further information on research design is available in the Nature Portfolio Reporting Summary linked to this article.

## Data availability

TCGA-BRCA, CCLE, and mouse data used in this article before and after integration can be downloaded at https://webshare.bioinf.unc.edu/public/baprice/AJIVE_jDR_Integration/. All relevant processed data used in figures in this study and supplementary information files can be downloaded at https://webshare.bioinf.unc.edu/public/baprice/AJIVE_jDR_Integration/FigureData. Processed data for eight other TCGA subtypes (TCGA-BLCA, TCGA-COAD, TCGA-GBM, TCGA-KIRC, TCGA-LUAD, TCGA-LUSC, TCGA-READ, TCGA-SKCM) are also provided at the link. Remaining clinical data are available as noted in the article or in the jupyter notebook upon reasonable request, and according to data use guidelines of each particular data set.

## Code availability

Step-by-step code to perform AJIVE integration and calculate gene joint statistics are available as Jupyter notebooks at https://github.com/baprice/AJIVE-jDR-Integration. https://doi.org/10.5281/zenodo.7309384.

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

## Acknowledgements

This work was supported by funds from the NCI Breast SPORE program (P50-CA58223), NCI U01-CA238475, the Breast Cancer Research Foundation, and the Susan G. Komen (SAC-160074). We thank Iain Carmichael for technical assistance in using the py_jive package and Dan Hollern for consultation on the GEMMs dataset. The results published here are in whole or part based upon data from the Cancer Genome Atlas managed by the NCI and NHGRI (dbGaP accession phs000178). These studies also included data from CALGB 40601: Randomized Phase III Trial of Paclitaxel Combined With Trastuzumab, Lapatinib, or Both As Neoadjuvant Treatment of HER2-Positive Primary Breast Cancer (dbGaP Study Accession: phs001570.v2.p1).

## Author contributions

B.A.P. and J.S.P. conceived the project. L.E.M. acquired and processed raw sequencing data. B.A.P. performed data integration and conducted data analysis. J.S.P., C.M.P., and J.S.M., assisted in the interpretation of results. B.A.P. wrote the manuscript with input from J.S.P., C.M.P., and J.S.M. All authors read and approved the final manuscript.

## Competing interests

The authors declare no competing interests.
