## [Peer Review File · Communications Biology]

Reviewers' comments:

Reviewer #1 (Remarks to the Author):

Authors have extensively addressed my comments.

Moderate/minor comment: It is interesting that tumor purity shows correlation with gene variation. Could authors elaborate their conclusions on this analysis? What we, as a scientific community, should be careful about and look for while performing data analysis? Perhaps this could be added to the discussion.

Reviewer #2 (Remarks to the Author):

The authors presented a joint dimension reduction method to integrate different data sources based on the same features (genes). To show that integration can capture joint-acting variation, the authors applied the model to CCLE and TCGA data and showcased how joint variances and translatable genes in breast cancer can be determined. Through out-of-sample validation using clinical trial data, the authors showed AJIVE can improve drug response prediction among patients. The authors have addressed most main concerns, and here are some remaining ones.

1. The potential application could be restricted by availability of data in different cancers. Breast cancer has a tremendous representation in CCLE, but this is not the case for other types of cancers. Could the authors comment on use of pan-cancer cell lines for joint dimension reduction to predict drug responses?
2. It seems that the potential scenario to apply this method is somewhat dependent on previous knowledge or findings. For example, the authors applied AJIVE to show that responses to lapatinib was improved. However, this does not showcase how this method can be used to quickly generate hypotheses as in perspective analysis.
3. In the previous review, a simulation test were requested and the authors responded that since the example they gave was tested under different sample sizes, simulation is not necessary. The sample size is only one of the many issues that can affect the model performance. Batch effect in which separate batches maybe coming from each level of data to be integrated (eg., different batch in patient tumor samples). Moreover, the range of samples they tested is between 50 to 935. How about less than 50? How about integrating more than 2 datasets?
4. Regarding the utility of individual level data: The authors pointed out the presence of system-specific variation. But they fell short in examining how to utilize these information to improve the drug response prediction, especially given the modest performance of the current model in comparison to other existing models.
5. The authors preprocessed expression data using the same pipeline to control for unnecessary variation. While this is a good practice, in case where raw data is inaccessible, would the method perform similarly? Would it be appropriate to apply the method on preprocessed units, e.g., FPKM, TPM etc.? If so, is there any additional processing needed (e.g., standardization)?
6. How are this SVD-based method differing from an NMF-based method where the common factor matrix W (assuming $X \approx WH, X \in \mathbb{R}^{(p_{\text{gene}} \times n_{\text{sample}})}$) were extracted?

Minor concerns:

The input data is gene expression/RNAseq data for all examples in the manuscript. Whether this method is applicable to other -omic data is not proven. Therefore, it can be misleading to title it "...genomic findings..." please revise the title to reflect the work as transcriptomic data rather than genomic data.

Again in the title, the manuscript only demonstrated the application of this model in the cancer setting. The title can be infer to much broader context. Please be specific.

In Figure 2C&D&E&F it is difficult to see where breast cancer cell lines lie in. Consider adding

annotations or using clearer labeling.

Reviewer #3 (Remarks to the Author):

We thank the authors for this revised version of their manuscript. On the whole, we found it easier to read this time, and we think the paper has improved. For most of the issues, the authors have made a satisfactory adjustment / correction. We have only a few remaining questions and comments. Overall, our main remaining concern is related to the code and data availability. Indeed, only the code corresponding to the first part of the analysis is available from GitHub, without detailed documentation.

A few numbers of our comments are not answered in the rebuttal letter we still think some information about the biological interpretation of the right singular vectors would be helpful - this is crucial to understanding the benefit of using the joint structure regarding the formula for the joint statistic - we think it still needs a subscript for the gene as well as the J and I subscripts. We do not really understand the answer to this comment in the rebuttal. Figure S4 (now Figure S5): If the plots were consolidated by row then comparisons of joint vs original would be easier.

Regarding the revised version of the manuscript

"Out-of-sample" and "mode" would require better definitions in the abstract/introduction

There remain some synonyms that can be confusing: cohort-specific versus systems-specific

I'm not sure how to understand this sentence "Furthermore, we observed that molecular subtype differentiation remained present when clustering using breast cell lines (Supplementary Fig. S3b), indicating there is primary tumor specific subtype variation that is not being captured by the breast cell lines. ". If it remains present, why conclude that it's not being captured?

TRANSACT is not detailed, why applied only to the cell lines and not the gemm? Would need some detail on the method and how different it is, as it currently just states that it's non-linear and not applied on out-of-sample data.

Elastic net models and code do not seem to be available from the GitHub repo. Overall, not all the information to reproduce the analyses is available from GitHub/jupyter notebook. Some data might be protected, but the code at least could be available.

Typos legend fig 2 => cell lines*

Response to Reviewers

Reviewers' comments:

Reviewer #1 (Remarks to the Author):

Authors have extensively addressed my comments.

Moderate/minor comment: It is interesting that tumor purity shows correlation with gene variation. Could authors elaborate their conclusions on this analysis? What we, as a scientific community, should be careful about and look for while performing data analysis? Perhaps this could be added to the discussion.

Thank you for bringing tumor purity to our attention. We interpret the correlation as a result of using integration to capture only the translatable aspects of both high and low purity level samples. This suggests that integration can be used to remove the confounding effects of tumor purity on transcriptomic analyses.

We have included the following in our revised manuscript on line 299:

“Furthermore, we found TCGA-BRCA sample purity correlated with the amount of joint variation, indicating that jDR is capturing the translatable aspects of both high and low purity samples. As a result, we hypothesize that jDR integration could be used help remove the confounding effects of purity level from an analysis [33].”

Reference

33. Aran, D., Sirota, M. & Butte, A. J. Systematic pan-cancer analysis of tumour purity. *Nat Commun* 6, 8971 (2015).

Reviewer #2 (Remarks to the Author):

The authors presented a joint dimension reduction method to integrate different data sources based on the same features (genes). To show that integration can capture joint-acting variation, the authors applied the model to CCLE and TCGA data and showcased how joint variances and translatable genes in breast cancer can be determined. Through out-of-sample validation using clinical trial data, the authors showed AJIVE can improve drug response prediction among patients. The authors have addressed most main concerns, and here are some remaining ones.

We thank the reviewer for their comments and extensive understanding of our work.

1. The potential application could be restricted by availability of data in different cancers. Breast cancer has a tremendous representation in CCLE, but this is not the case for other types of cancers. Could the

authors comment on use of pan-cancer cell lines for joint dimension reduction to predict drug responses?

We agree with the reviewer that our work presents a best-case scenario where you have access to a dataset that well represents your target clinical population. However, as shown in Supplemental Fig. S1, joint dimension reduction can identify the relevant subpopulations from cell lines irrespective of the number of cell lines for a subtype in CCLE. To better illustrate this, the number of cell lines for each type have been included as a table in Fig. S1I.

Number of cell lines			
LUNG	173	LIVER	25
LARGE_INTESTINE	54	URINARY_TRACT	25
BREAST	50	KIDNEY	23
CENTRAL_NERVOUS_SYSTEM	49	SOFT_TISSUE	17
SKIN	47	BONE	16
OVARY	45	AUTONOMIC_GANGLIA	15
PANCREAS	41	THYROID	11
STOMACH	37	PLEURA	9
FIBROBLAST	34	BILIARY_TRACT	7
UPPER_AERODIGESTIVE_TRACT	30	PROSTATE	7
ENDOMETRIUM	28	SALIVARY_GLAND	2
OESOPHAGUS	27	SMALL_INTESTINE	1

Supplemental Figure S1I

Specific to drug response prediction, we demonstrate the use of pan-cancer cell lines in Supplemental Fig. S4 and found that including pan-cancer cell lines no longer significantly improved our predictive model over using only breast cell lines. However, as in Supplemental Fig. S1, jDR integration could be used to identify subpopulations of pan-cancer datasets to improve the specificity of the integrated dataset. In the revised manuscript, we note this in the discussion on lines 292-299:

“Although performance remained the same after jDR integration with TCGA-BRCA, the choice of integrated dataset had a significant effect on the model’s predictive performance and could potentially be improved through selecting datasets that better represent both the model system population and clinical population. In scenarios where representative datasets are not available, our findings using jDR to integrate pan-cancer cell lines from CCLE and all tumor types of TCGA (Supplemental Fig. S4) suggest that jDR integration could be used to identify relevant subpopulations within large public data repositories.”

2. It seems that the potential scenario to apply this method is somewhat dependent on previous knowledge or findings. For example, the authors applied AJIVE to show that responses to lapatinib was improved. However, this does not showcase how this method can be used to quickly generate hypotheses as in perspective analysis.

The reviewer raises an excellent question on how our work can be applied to a prospective analysis.

The focus of our paper is on improving the translation of model system transcriptomic analysis to clinical settings. To do this, we must first establish that the use of joint variation is more meaningful for translation than using the entire data. As a result, in order to best demonstrate the improvement of using the joint variation, we must use previous findings that have associated clinical data. Our approach is not dependent on previous findings, but we use them to directly show a clinical improvement.

Both *TRANSACT* [1] and *Celligner* [2] articles do an excellent job of demonstrating how integration can be used to generate hypotheses in an exploratory, or prospective, analysis and we did not wish to reiterate this in our work. *TRANSACT* demonstrates use of generating hypothesis from predictive modeling in a drug screening scenario and *Celligner* demonstrates how identification of joint-acting samples can benefit clustering and gene set analysis. Both of these articles suggested how this could lead to improvements in clinical translation, but did not show showcase it. In our work, while we do demonstrate how clustering (Fig. 2C-F) and gene set analysis (Fig. 3C-F) benefit from using joint variation, we instead focus directly on how the clinical translation is improved in predictive modeling (Fig. 4B-C, Fig. 5D-E) and biomarker identification (Fig. 5A-C).

In the revised manuscript, we have clarified this on lines 65-72:

“Specific to translating cell line or mouse variation to human datasets, methods such as Celligner [14], PRECISE [15], and TRANSACT [16] (a non-linear implementation of PRECISE) have shown success in clustering model system and tumor biopsy pairings. These methods have successfully shown how integration can be used to generate more informed hypotheses from model system datasets. However, only TRANSACT has demonstrated an improvement in drug response prediction, but these predictive models have not been validated on clinical datasets outside of the integration (i.e. out-of-sample validation). As a result, the ability for jDR to be used for clinical translation remains uncertain.”

References:

1. Jang, I. S., Neto, E. C., Guinney, J., Friend, S. H. & Margolin, A. A. SYSTEMATIC ASSESSMENT OF ANALYTICAL METHODS FOR DRUG SENSITIVITY PREDICTION FROM CANCER CELL LINE DATA. *Pac Symp Biocomput* 63–74 (2014).
2. Warren, A. et al. Global computational alignment of tumor and cell line transcriptional profiles. *Nat Commun* 12, 22 (2021).
3. In the previous review, a simulation test were requested and the authors responded that since the example they gave was tested under different sample sizes, simulation is not necessary. The sample size is only one of the many issues that can affect the model performance. Batch effect in which separate batches maybe coming from each level of data to be integrated (eg., different batch in patient tumor samples). Moreover, the range of samples they tested is between 50 to 935. How about less than 50? How about integrating more than 2 datasets?

We agree with the reviewer that there are many factors that can affect model performance. We misinterpreted the previous review response to be in reference to the integration aspect of our work, not the predictive modeling. We have now included in our revised manuscript results of simulating smaller cell line counts in our ERBB2-targeting response modeling experiment as Supplemental Fig. S7. We show that increasing the number of relevant cell lines increases the amount of captured joint variation, leading to improved predictive modeling using the joint matrix.

Supplemental Figure S7

In the revised manuscript, we have added the following on lines 230-235:

“To test the extent to which integration improves the success of the models, a retrospective analysis was performed bootstrapping the AJIVE integration and elastic net modeling procedures at varying breast cell line sample sizes (Supplementary Fig. S7). By adding more breast cell lines to the integration, the amount captured joint variation was increased, leading to improved predictive model performance using the joint matrix.”

Although it is possible to integrate more than 2 datasets using AJIVE, the interpretation of individual level variation becomes more challenging as you have to consider not only the variation specific to each input matrix, but also every combination of input matrices (i.e. “partial-joint” or “partial-individual” variation). As a result, we did not include such an analysis in our work. However, this is an area of interest in future work.

References:

1. Feng, Q., Jiang, M., Hannig, J. & Marron, J. S. Angle-based joint and individual variation explained. *Journal of Multivariate Analysis* **166**, 241–265 (2018).

4. Regarding the utility of individual level data: The authors pointed out the presence of system-specific variation. But they fell short in examining how to utilize these information to improve the drug response prediction, especially given the modest performance of the current model in comparison to other existing models.

We agree with the reviewer that there is likely some amount of variation in the individual level data. However, as we demonstrate in Figures 4b, 4c, 5d, and Supplemental Figures S4c, S4e, and S5c, predictive models trained on this individual level data do not perform above chance. This indicates that the majority of variation in the individual matrices is not translatable and should not be considered when training clinical predictive models from model systems.

5. The authors preprocessed expression data using the same pipeline to control for unnecessary variation. While this is a good practice, in case where raw data is inaccessible, would the method perform similarly? Would it be appropriate to apply the method on preprocessed units, e.g., FPKM, TPM etc.? If so, is there any additional processing needed (e.g., standardization)?

AJIVE is a scale-free method that packs scaling data into the diagonal matrices of the SVD steps. This scaling data is not considered in the subsequent principal angle analysis. Thus, we hypothesize it will work on datasets that have been normalized differently. However, it is our current recommendation that the datasets be processed identically whenever possible to control for any additional batch effects as the reviewer notes.

Processing steps are noted on lines 330-332

“Genes with no expression in any dataset were removed and an intersecting gene list across datasets was used. Genes were mean-centered as required by AJIVE.”

6. How are this SVD-based method differing from an NMF-based method where the common factor matrix W (assuming $X \approx WH, X \in \mathbb{R}^{(p_{\text{gene}} \times n_{\text{sample}})})$ were extracted?

The main difference between AJIVE and NMF-based methods is the identification of individual-level variation. NMF-based methods such as intNMF [1] will only identify common (i.e. shared) factors that performs best for clustering tasks [2].

References:

1. Chalise, P. & Fridley, B. L. Integrative clustering of multi-level 'omic data based on non-negative matrix factorization algorithm. PLoS ONE 12, e0176278 (2017).
2. Cantini, L. et al. Benchmarking joint multi-omics dimensionality reduction approaches for the study of cancer. Nat. Commun. 12, 124 (2021).

Minor concerns:

The input data is gene expression/RNAseq data for all examples in the manuscript. Whether this method is applicable to other -omic data is not proven. Therefore, it can be misleading to title it "...genomic findings..." please revise the title to reflect the work as transcriptomic data rather than genomic data. Again in the title, the manuscript only demonstrated the application of this model in the cancer setting. The title can be infer to much broader context. Please be specific.

The title has been changed to "Translating transcriptomic findings from cancer model systems to humans through joint dimension reduction"

In Figure 2C&D&E&F it is difficult to see where breast cancer cell lines lie in. Consider adding annotations or using clearer labeling.

Figures 2E and 2F identify where the breast cancer cell lines are compared to all cell lines in Figures 2C and 2D.

Reviewer #3 (Remarks to the Author):

We thank the authors for this revised version of their manuscript. On the whole, we found it easier to read this time, and we think the paper has improved. For most of the issues, the authors have made a satisfactory adjustment / correction. We have only a few remaining questions and comments. Overall, our main remaining concern is related to the code and data availability. Indeed, only the code corresponding to the first part of the analysis is available from GitHub, without detailed documentation.

Thank you for your comments and insight. We have updated and organized the GitHub repository with all code used to generate the figures in the manuscript including code for training the elastic net models. More detailed documentation has been included within each jupyter notebooks to help in the step-by-step analysis.

A few numbers of our comments are not answered in the rebuttal letter we still think some information about the biological interpretation of the right singular vectors would be helpful - this is crucial to understanding the benefit of using the joint structure

We greatly appreciate the reviewer's understanding and intuition behind our approach.

In the revised manuscript, we have included the following on line 377:

"By integrating the datasets across genes, the right singular vectors of the decomposition now represent ranks of "metagenes" across the datasets. Each metagene contains partial combinations of genes that AJIVE then compares and identifies as joint or individually behaving."

Regarding the formula for the joint statistic - we think it still needs a subscript for the gene as well as the J and I subscripts. We do not really understand the answer to this comment in the rebuttal.

Subscripts have been added to the joint statistic equation. The description has been updated on line 396:

“We developed a statistic to determine the “joint behavior” of a gene. This is represented by the log-ratio of each gene g , variance in the joint matrix σ_j^2 by the variance in the individual matrix σ_I^2 .”

$$\log \left(\frac{\max(\sigma_{J_g}^2, s)}{\max(\sigma_{I_g}^2, s)} \right)$$

Figure S4 (now Figure S5): If the plots were consolidated by row then comparisons of joint vs original would be easier.

Supplemental Figure S5 has been updated.

Supplemental Figure S5

Regarding the revised version of the manuscript

“Out-of-sample” and “mode” would require better definitions in the abstract/introduction

“Out-of-sample” has been better defined on line 69-72:

“However, only TRANSACT has demonstrated an improvement in drug response prediction, but these predictive models have not been validated on clinical datasets outside of the integration (i.e. out-of-sample validation). As a result, the ability for jDR to be used for clinical translation remains uncertain.”

In the abstract, “modes of genomic variation” has been changed to “aspects of genomic variation” on line 30. On line 58, “determine shared modes of variation” has been changed to “determine shared components of variation.”

There remain some synonyms that can be confusing: cohort-specific versus systems-specific

Instances of “system-specific” have been changed to “cohort-specific” to prevent confusion.

I’m not sure how to understand this sentence “Furthermore, we observed that molecular subtype differentiation remained present when clustering using breast cell lines (Supplementary Fig. S3b), indicating there is primary tumor specific subtype variation that is not being captured by the breast cell lines. “. If it remains present, why conclude that it’s not being captured?

We agree with the reviewer that this sentence is confusing. The aspect of molecular subtype variation noted here is not being captured by the cell lines because we observe the clustering in the **individual** variation that is specific to TCGA-BRCA ($I_{\text{TCGA-BRCA}}$). This distinction was not clear in the text. Lines 115-119 have been revised:

“Furthermore, we observed that tumors continued to cluster according to molecular subtype in the individual variation (Supplementary Fig. S3b). This indicates that there are aspects of primary tumor molecular subtype variation that act specific to TCGA-BRCA and are not being captured by the breast cell lines of CCLE. Similar clustering effects were observed when integrating pan-cancer TCGA samples with CCLE (Supplementary Fig. S3c-e).”

TRANSACT is not detailed, why applied only to the cell lines and not the gemm? Would need some detail on the method and how different it is, as it currently just states that it’s non-linear and not applied on out-of-sample data.

In the revised manuscript, a comparison to TRANSACT has been included in Fig. 5D. A description of TRANSACT has been updated and detailed on lines 437-445:

“TRANSACT [28] is a non-linear implementation of PRECISE [29] that utilizes kernel-PCA to derive non-linear principal components (NLPCs) that are then geometrically compared into principal vectors. Vectors that best balance the contributions of the input datasets are called *consensus*

features and are used to project the datasets into a reduced-dimension joint space. To compare with our AJIVE approach, we repeated our experiment integrating CCLE-BRCA and TCGA-BRCA using TRANSACT and applied the trained predictive model on CALGB 40601. Code to run the integration was used from the TRANSACT package: <https://github.com/NKI-CCB/TRANSACT>. TRANSACT elastic net models were trained on consensus features using the same modeling procedure as detailed above.”

Updated Figure 5D

Elastic net models and code do not seem to be available from the GitHub repo. Overall, not all the information to reproduce the analyses is available from GitHub/jupyter notebook. Some data might be protected, but the code at least could be available.

The GitHub repository has been updated with clearer documentation and includes elastic net model training code.

Typos legend fig 2 => cell liens*

Fixed

REVIEWERS' COMMENTS:

Reviewer #2 (Remarks to the Author):

The authors addressed our questions well; a few major issues, including perspective analysis and drug prediction, were deemed by the authors to be somewhat out of the scope of their manuscript. Here's some minor comments:

1. Suppl. Figure 3. No explanation for panel E was given.
2. Fig 2 C-D legend: "breast" is listed twice.
3. Fig 4B and 5D both show the ROC curve of prediction performances for pCR. However, the legend is inconsistent. For example, "original" was red in 4B but blue in 5D. This is confusing.

Reviewer #3 (Remarks to the Author):

The authors answered all our comments